# Bi-Directional Interactions between Glucose-Lowering Medications and Gut Microbiome in Patients with Type 2 Diabetes Mellitus: A Systematic Review

**DOI:** 10.3390/genes14081572

**Published:** 2023-08-01

**Authors:** Ruolin Li, Fereshteh Shokri, Alejandro Lopez Rincon, Fernando Rivadeneira, Carolina Medina-Gomez, Fariba Ahmadizar

**Affiliations:** 1Department of Internal Medicine, Erasmus University Medical Center, 3015 GD Rotterdam, The Netherlands; r.li@erasmusmc.nl (R.L.); f.rivadeneira@erasmusmc.nl (F.R.); m.medinagomez@erasmusmc.nl (C.M.-G.); 2Department of Epidemiology, Erasmus University Medical Center, 3015 GD Rotterdam, The Netherlands; fereshteh.m.shokri@gmail.com; 3Department of Data Science & Biostatistics, Julius Global Health, University Medical Center Utrecht, 3508 GA Utrecht, The Netherlands; alejandro.lopezrn@hotmail.com

**Keywords:** gut microbiome, glucose-lowering medications, type 2 diabetes mellitus

## Abstract

*Background*: Although common drugs for treating type 2 diabetes (T2D) are widely used, their therapeutic effects vary greatly. The interaction between the gut microbiome and glucose-lowering drugs is one of the main contributors to the variability in T2D progression and response to therapy. On the one hand, glucose-lowering drugs can alter gut microbiome components. On the other hand, specific gut microbiota can influence glycemic control as the therapeutic effects of these drugs. Therefore, this systematic review assesses the bi-directional relationships between common glucose-lowering drugs and gut microbiome profiles. *Methods:* A systematic search of Embase, Web of Science, PubMed, and Google Scholar databases was performed. Observational studies and randomised controlled trials (RCTs), published from inception to July 2023, comprising T2D patients and investigating bi-directional interactions between glucose-lowering drugs and gut microbiome, were included. *Results:* Summarised findings indicated that glucose-lowering drugs could increase metabolic-healthy promoting taxa (e.g., *Bifidobacterium)* and decrease harmful taxa (e.g., *Bacteroides* and *Intestinibacter*). Our findings also showed a significantly different abundance of gut microbiome taxa (e.g., *Enterococcus faecium* (i.e., *E. faecium*)) in T2D patients with poor compared to optimal glycemic control. *Conclusions:* This review provides evidence for glucose-lowering drug and gut microbiome interactions, highlighting the potential of gut microbiome modulators as co-adjuvants for T2D treatment.

## 1. Introduction

The human gut microbiome is composed of a collection of microorganisms (namely archaea, eukaryotes, viruses, and bacteria) that reside in humans’ digestive tract and participate in different biological processes. In recent years, accumulating evidence supports a significant role of the gut microbiome in the aetiology and physiopathology of type 2 diabetes mellitus (T2D) [1,2,3,4,5]. Alterations in the composition and activity of the gut microbiome (known as intestinal dysbiosis) can reduce short-chain fatty acid (SCFA) synthesis [6], which could damage the gut barrier integrity, influence pancreatic β-cell proliferation, and lessen insulin biosynthesis, thus boosting a rapid progression toward insulin resistance in T2D [7]. The gut microbiome has an innate mouldable nature, so it is a feasible therapeutic target for T2D. The human gut microbiome composition is easily disrupted by external conditions such as the surrounding environment, the place of residence, drug and food interventions, and lifestyle factors [8].

The gut microbiome is crucial in drug metabolism, which is complicated and elusive. There are many pathways by which the gut microbiome modulates drug metabolism [9], for example, by direct secretion of drug-metabolising enzymes in the intestine, competition for receptors or transporters in host tissues with drugs via the production of bacterial metabolite, and microbial modulation on the activity of drug-metabolising enzymes in host tissues [10]. In addition, some drugs are metabolised by the intestinal microbiota into specific metabolites that cannot be formed in the liver; the gut microbiota can also alter the systemic bioavailability of certain drugs prior to absorption [11].

The interaction between gut microbes and commonly used drugs in T2D patients is complex and can be bi-directional [12]. On the one hand, enzyme activity in gut microbial communities can alter the structure of drugs, thereby influencing their bioavailability. In particular, variability in the component and metabolic capacity of the gut microbiome can significantly influence the toxicity and clinical efficacy of drugs. On the other hand, drugs can also alter the composition and function of the gut microbial community, thereby changing the intestinal microenvironment and affecting microbial metabolism. The interactions between gut microbiome and drugs are summarised in Figure 1, as published previously [13]. However, despite numerous studies on these relationships, findings still need to be more conclusive. Even recently, a review addressing this bi-directional association was published [12], however, it used a different methodology and presented limitations that merit consideration in a new study. Therefore, in this study, we systematically reviewed and summarised the current evidence from observational studies and randomised control trials (RCTs) on the bi-directional interaction between glucose-lowering medications and gut microbiome in T2D patients.

## 2. Methods

### 2.1. Search Strategy for the Systematic Review

We reviewed published studies that evaluated the association between glucose-lowering medications and the gut microbiome in T2D patients. We followed the “Preferred Reporting Items for Systematic Review and Meta-Analyses” (PRISMA) guidelines for our study [14]. All observational studies, including descriptive, cross-sectional, case-control, cohort, or case-cohort human studies and RCTs that evaluated the gut microbiome of adults with T2D were assessed from inception to July 2023. For RCTs, studies with two-arm or multiple-arm study designs were included. We searched for eligible studies in Embase, Web of Science, PubMed, and Google Scholar. The search strategy was conducted using MeSH terms for [*glucose-lowering medications*] AND [*gut microbiome*] AND [*T2D patients*]. The search terms for each database can be found in Appendix A.

### 2.2. Inclusion and Exclusion Criterion

Two investigators (RL and FS) screened independently to determine the study’s eligibility. Discrepancies in study selection were resolved through the consensus and consulting of a third reviewer (FA). We identified 9268 unique articles. After the first screening based on the title and abstract, 159 articles remained. After reviewing the full texts, 17 articles [15,16,17,18,19,20,21,22,23,24,25,26,27,28,29,30,31] met the full inclusion criteria (Figure 2). We excluded studies that (i) were conducted on animal models or those based on culture techniques, (ii) did not use 16s rRNA sequencing or shotgun sequencing for microbiome quantification, (iii) investigated the effects of environment or diet or physical activity or other irrelevant drugs on the gut microbiome of patients with T2D, (iv) evaluated oral or skin microbiome, and (v) focused on the side effects of the drugs rather than on their therapeutic effects.

### 2.3. Data Extraction

Two investigators (RL and FS) independently extracted all essential information from the selected studies. This information included study characteristics, such as the year of publication, authors, study design, number of participants, drug dosages, follow-up period, results reported, bioinformatic pipeline for microbiome sequencing data processing, reference genome for taxa mapping, and methods used for differentially abundant taxa identification. The extracted information is summarised in Appendix A. No synthesis or statistical data analysis was performed.

### 2.4. Quality and Risk of Bias Assessment

Two investigators (RL and FS) independently carried out the quality assessments of the 17 included articles (19 studies) according to the Cochrane tools for assessing the risk of bias in randomised trials (RoB 2 tool) [32] and non-randomised studies of interventions (ROBINS-I) [33]. We excluded three articles due to low quality: One [28] had no clear description of the trial design and included a very small sample size (*n* = 8); the other [27] reported differences before the treatment and had no accurate conclusion about the relationship; and another one [31] had only three T2D patients included in the treatment group [31]. Therefore, the final number of articles included in our review is 14 (18 studies) [15,16,17,18,19,20,21,22,23,24,25,26,29,30] as listed in Appendix A (with 4 articles each containing studies of two types of glucose-lowering drugs)**.**

## 3. Results

Appendix A listed details of 10 observational studies selected to study “the effects of glucose-lowering medication on gut microbiome composition”. Appendix A listed details of eight RCTs selected to study “the effects of glucose-lowering medication on gut microbiome composition”, and Appendix A listed details of six studies (both observational studies and RCTs) selected to study “the effects of the gut microbiome on glycemic control”. The characteristics of all selected studies and risk-of-bias evaluation results are summarised in Table 1.

The effects of six types of glucose-lowering medications on the gut microbiome composition were reported. These medications included: (1) metformin; (2) sulfonylureas; (3) α-glucosidase inhibitor (α-GI); (4) glucagon-like peptide 1 receptor agonist (GLP-1RA); (5) sodium-glucose cotransporter-2 (SGLT2) Inhibitor; and (6) dipeptidyl peptidase 4 inhibitor (DPP-4 inhibitor). The metformin’s effect was investigated in seven observational studies [15,16,17,18,19,20,21] and two RCTs [22,23]. The sulfonylureas’ effect was investigated in two RCTs [24,26], whereas the effect of GLP-1RA was reported in one observational study [25]. The α-GI’s effect was reported in two observational studies [15,29] and two RCTs [24,30]; the SGLT2 Inhibitor was reported in one RCT [26] and DPP-4 inhibitor was reported in one RCT [30]. Among these studies, 10 [15,17,19,21,22,25,26,29] used 16S rRNA sequencing and eight [16,18,20,23,24,30] used metagenomic sequencing. The sample size of these studies ranged from 14 to 106 individuals. A self-controlled design, in which intervention subjects served as their controls, with comparison before and after treatment, was used in seven observational studies [18,19,20,21,29] and all RCTs, except one [23] that used a placebo control group. Eleven of these studies were conducted in East Asian countries [15,18,21,22,24,25,29,30], six in European countries [16,19,20,23,26], and one in Colombia [17].

### 3.1. Effects of Glucose-Lowering Medication on Gut Microbiome Composition

The effects of glucose-lowering medication on gut microbiome composition are summarised in Table 2.

#### 3.1.1. Metformin

Metformin is the most frequently prescribed medication among biguanides to treat T2D, with beneficial effects on blood glucose and cardiovascular mortality [34,35]. Although the widely known mechanism of metformin action involves the suppression of hepatic gluconeogenesis, thereby reducing glucose production, new findings indicate that the gut microbiome might mediate the metformin effect [36,37,38,39,40].

##### Results from Observational Studies

As shown in Table 2, seven studies evaluated the effects of metformin on the gut microbiome composition changes, from which five also evaluated changes in microbiome diversity.

Among the seven studies, four used the same individuals to evaluate the treatment effect (i.e., the gut microbiome of the same individual was measured and compared at baseline and after intervention). One study [18] by Sun et al. assessed the gut microbiome profile of 30 newly diagnosed T2D patients naively treated with 1000 mg/day of metformin for three days. After treatment, patients showed a lower abundance of *B. fragilis* (adjusted *p* value = 0.0002) and decreased α diversity (measured by the Shannon index) (*p* < 0.0001). The other study [19] by Napolitano et al. evaluated the gut microbiome of 14 T2D patients treated with metformin (1000 mg/day) for over three months and found that patients after treatment showed different abundances in four genera, including *Adlercreutzia*, *Firmicute (other)*, *Eubacterium*, and *SMB53*, which did not endure multiple testing corrections (adjusted *p* > 0.05). Another study [20] by Elbere et al. evaluated gut microbiome alterations in 50 newly diagnosed T2D patients treated with metformin (endocrinologist individually determined dosage) for seven days. This study found that patients after treatment showed changes in 26 taxa after adjusting for baseline HbA1c levels, including *Bifidobacteriales*, *Bifidobacteriaceae*, *Bifidobacterium*, and seven other taxa which increased in abundance, and Clostridiaceae, *Bacteroides_vulgatus*, *Bacteroides_vulgatus_unclassified*, and 13 other taxa which decreased (adjusted *p* < 0.05) (see Table 2). However, no significant changes in α diversity were detected (*p* > 0.05). The last study [21] by Nakajima et al. assessed gut microbiome changes from 21 T2D patients not using additional medications affecting the gut microbiome (i.e., α-glucosidase inhibitors, dipeptidyl peptidase IV inhibitors, Proton-pump inhibitors (PPIs), and H2 blockers) and who took metformin for four weeks (metformin dosage was not specified). This study found that the abundance of phylum *Firmicutes* and the ratio of the *Firmicutes* and *Bacteroidetes* abundances decreased (*p* < 0.05). Metformin did not significantly affect α diversity (*p* > 0.05).

In addition to the above-mentioned studies, three studies compared microbiome profiles of T2D individuals taking medication with others who did not. One study was conducted by Cuesta-Zuluaga et al. [17], in which the gut microbiome composition was compared between T2D patients with (*n* = 14) and without (*n* = 14) metformin treatment (dosage and duration of therapy were not specified). All participants were matched by age, sex, and body mass index (BMI). This study found that metformin-treated patients had a higher abundance of *Prevotella* and *Megasphaera* and a lower abundance of Oscillospira, Barnesiellaceae, and *Clostridiaceae 02d06* compared to the untreated group (adjusted *p* < 0.05). Differences in α diversity were not reported. In the other study conducted by Forslund et al. [16], T2D patients (*n* = 93) treated with metformin (unspecified dosage and duration) had an increased abundance of *Escherichia* spp. and a decreased abundance of *Intestinibacter* spp. compared to the untreated group (*n*  =  106). The results were still significant after correction for different covariates, including sex, BMI, fasting plasma glucose levels, and serum insulin (all *p* < 1 × 10^−8^). However, the adjustment for covariates was made one at a time and not altogether during follow-up. Conversely, no differences in α diversity (richness) were identified across the two groups (*p* > 0.05). In another study by Zhang et al. [15], T2D patients following metformin treatment (*n* = 51, duration > 3 months, unspecified dosage information) presented a decreased abundance of Erysipelotrichi and *Ruminococcus*, and an increased abundance of Spirochaete (adjusted *p* < 0.01) and *Turicibacter* (adjusted *p* < 0.01) compared to the non-therapeutic group (*n* = 26). The α diversity was not affected by metformin treatment. However, this study did not correct for any confounder, and the risk of potential confounding bias needs to be evaluated critically.

##### Results from RCTs

As shown in Table 2, two RCTs [22,23] assessed the effects of metformin on the gut microbiome. One RCT was performed by Tong et al. [22] on 100 T2D patients (*n* = 100) following a 12-week metformin treatment (unspecified dosage). This study found that after treatment, compared to their baseline gut microbiome, the abundance of *Clostridium XIVa*, *Erysipelotrichaceae incertae sedis*, *Escherichia-Shigella*, *Fusobacterium*, *Flavonifractor*, *Lachnospiraceae*, *Clostridium* XVIII and IV, *Blautia* spp., and *Anaerostipes* increased, while the abundance of *Bacteroides*, *Parabacteroides*, *Alistipes*, *Oscillibacter*, and *Ruminococcaceae* decreased (adjusted *p* < 0.05). The α diversity (Shannon index) also increased with treatment (p < 0.05). The other RCT was performed by Wu et al. [23], which included 40 treatment-naïve, recently diagnosed T2D patients (placebo n = 18, and metformin n = 22) for a 4-month treatment (1700 mg/day). This study found that after treatment, in the metformin group, the abundance of *γ-Proteobacteria* increased, and the abundance of *Firmicutes* and *Intestinibacter* decreased compared to the placebo group (adjusted *p* < 0.05). In addition, the authors obtained similar results in a subset of the placebo group (*n* = 13) that switched to metformin (850 or 1700 mg/day) six months after the trial. Microbiome diversity was not evaluated in this study.

##### 3.1.2. α-Glucosidase Inhibitors (α-GI)

###### Results from Observational Studies

Two studies [15,29] were included in this category (Table 2). One study [29] involved 18 T2D patients treated with acarbose (150 or 300 mg/day) for four weeks. Compared to their baseline gut microbiome, at the phylum level, the abundance of Actinobacteria increased, and Bacteroidetes decreased after treatment (adjusted *p* < 0.05). At the Genus level, the abundance of *Bifidobacterium*, *Eubacterium*, *Megasphaera*, and *Lactobacillus* increased after treatment, and the abundance of *Bacteroides*, *Blautia*, *Prevotella*, *Clostridium*, *Phascolarctobacterium*, and *Lachnoclostridium* decreased (adjusted *p* < 0.05). The α diversity did not show significant changes (*p* > 0.05). The other study [15] involved 17 T2D patients who were treated with α-GI for longer than three months (unspecified dosage) and 26 T2D patients from the non-therapeutic group. Compared to the control group, patients from the α-GI group showed a higher abundance of *Bifidobacterium* and *Lactobacillus* (adjusted *p* < 0.01), commonly used as probiotics. The α diversity was not affected by the treatment (*p* > 0.05). However, there was no adjustment for any covariate in this analysis, which is a limitation of this study.

###### Results from RCTs

There were two RCTs [24,30] assessing the effects of α-GI (Acarbose) on the gut microbiome (Table 2). One RCT [24] enrolled 51 T2D patients and evaluated a glycaemic control and the gut microbiome composition before and after a 3-month-treatment of α-GI (unspecified dosage). Results showed that acarbose increased the abundance of *Lactobacillus* and *Bifidobacterium* and decreased the abundance of *Bacteroides*, *Alistipes*, and *Clostridium* (adjusted *p* < 0.01). The α diversity (Shannon index) decreased after treatment (*p* < 0.05). The other RCT [30] included 44 T2D patients and used an open-labelled trial of 6 months (dosage: starting at 50 mg TID and increasing to 100 mg TID daily in the third week). This study found that compared to baseline, after treatment, there were 15 genera, including *Bifidobacterium*, *Lactobacillus*, *Solobacterium*, et al. and 30 species increased in abundance; meanwhile,18 genera, *including Bacteroides*, *Roseburia*, *Alistipes*, et al. and 46 species decreased (adjusted *p* < 0.05) (see Appendix A). Similar to the other RCT [24], the α diversity (Shannon index) decreased after treatment (*p* < 0.05).

#### 3.1.3. Glucagon-Like Peptide-1 Receptor Agonist (GLP-1RA)

##### Results from Observational Studies

As shown in Table 2, one study [25], with a self-controlled design, examined the effects of GLP-1RA on 40 T2D patients treated with liraglutide (1.2 mg/day) for four months. This study found that after treatment, at the phylum level, the abundance of Fusobacteria decreased, and the abundance of Verrucomicrobia and Actinobacteria increased (*p* < 0.05). At the genus level, the abundance of nine genera decreased, including (1) *Acinetobacter* (*p* = 0.016); (2) *Oscillospira* (*p* = 0.013); (3) *Acidaminococcus* (*p* = 0.021); (4) *Succinatimonas* (*p* = 0.042); (5) *S24_7* (*p* = 0.008); (6) *Megamonas* (*p* = 0.0005); (7) *Alistipes* (*p* = 0.035); (8) *Fusobacterium* (*p* = 0.017); and (9) *Megasphaera* (*p* = 0.0002). Further, the abundance of three genera increased, including (1) *Collinsella* (*p* = 0.011), (2) *Akkermansia* (*p* = 0.002), and (3) *Clostridium* (*p* = 0.002). The richness (abundance-based coverage estimator, i.e., ACE and Chao1) reduced but no significant changes in the Shannon and Simpson indexes (*p* > 0.05).

#### 3.1.4. Sulfonylureas

##### Results from RCTs

As shown in Table 2, two RCTs [24,26] investigated sulfonylureas. One RCT [24] involved 43 T2D patients treated with Glipizide for three months (unspecified dosage information). However, no changes in gene richness or α diversity were detected following treatment compared to the baseline gut microbiome of the patients, nor were there any changes in microbial taxa (at the species level). The other study [26], a double-blind RCT of gliclazide (30 mg/day) treatment for 12 weeks (*n* = 20), also did not show changes either in gut microbiome α diversity or composition (adjusted *p* > 0.05) when comparing stool profiles at baseline and after treatment, suggesting that the microbiome did not mediate the drug’s effect.

#### 3.1.5. SGLT2 Inhibitor

##### Results from RCTs

One RCT [26] conducted by van Bommel et al. investigated an SGLT2 inhibitor, which involved 24 T2D patients who underwent treatment with dapagliflozin (10 mg/day) for 12 weeks. However, neither α diversity nor gut microbiome composition changed after treatment (adjusted *p* > 0.05). Despite these observations, glucose homeostasis improved after treatment, suggesting that the effect of dapagliflozin is unlikely to be mediated by the gut microbiome.

#### 3.1.6. Dipeptidyl Peptidase 4 Inhibitor

##### Results from RCTs

One RCT [30] conducted by Zhang et al. investigated a dipeptidyl peptidase 4 inhibitor (vildagliptin), which involved 48 T2D patients who underwent treatment with vildagliptin (50 mg BID) for 6 months. Compared to baseline, one genera *Bifidobacterium* and two species increased in abundance after treatment, and four genera (including *Paraprevotella*, *Fusobacterium*, *Parabacteroides*, and *Bacteroides*) and eight species decreased (adjusted *p* < 0.05) (see Appendix A). However, the α diversity (Shannon index) did not change after treatment (*p* > 0.05).

### 3.2. Effects of the Gut Microbiome on Glycemic Control

The results of the effects of the gut microbiome on glycemic control are summarised in Table 3.

#### 3.2.1. Metformin

##### Results from Observational Studies

As shown in Table 3, one study [20] was found under this category. In this study, the gut microbiome profile of responders (defined as HbA1c levels during three months of metformin therapy had decreased by >12.6 mmol/mol (1%) (adjusted *p* = 0.01)) was compared with non-responders following a 3-month metformin treatment (2 × 850 mg/day) on 46 newly diagnosed T2D patients. At baseline, the non-responders’ group presented a lower abundance of *Prevotella copri* and a higher abundance of *E. faecium*, *Lactococcus lactis*, *Odoribacter*, and *Dialister* (adjusted *p* < 0.05) species as compared to the responders’ group. Conversely, no differences were observed in the α diversity between responders and non-responders. These results suggest that these specific taxa might mediate metformin’s therapeutic effects.

##### Results from RCTs

As shown in Table 3, one study [22] performed RCT. In this study, the gut microbiome profile of 100 T2D patients before and after a 12-week treatment of metformin (unspecified dosage) was compared, which significantly alleviated hyperglycemia. Decreased taxa after metformin treatment included *Bacteroides*, *Parabacteroides*, *Alistipes*, *Oscillibacter*, and *Ruminococcaceae*, which were significantly correlated with reduced hyperglycemia (adjusted *p* < 0.05). In addition, these taxa were also negatively correlated with the homeostasis model assessment of β-cell function (HOMA-β) (adjusted *p* < 0.05). For the increased taxa after treatment, *Blautia* and *Anaerostipes* were negatively correlated with fasting blood glucose (FBG) and 2-h postprandial blood glucose (PBG); *Clostridium XIVa*, *Erysipelotrichaceae incertae sedis*, *Escherichia-Shigella*, *Fusobacterium*, *Flavonifractor*, *Lachnospiraceae*, and *Clostridium XVIII and IV* were negatively correlated with hemoglobin A1c (HbA1c) (adjusted *p* < 0.05). The correlation between α diversity and treatment effects was not reported.

#### 3.2.2. Other Drugs

##### α-Glucosidase Inhibitor and Dipeptidyl Peptidase 4 Inhibitor

One study [30] reported the RCTs of two drugs, an α-glucosidase inhibitor (acarbose) and a dipeptidyl peptidase 4 inhibitor (vildagliptin). For both drugs, the author reported that baseline *Barnesiella intestinihominis* and *Clostridium citroniae* were associated with a high response in the glucagon-like peptide-1 (GLP-1) levels (i.e., the percentage change in GLP-1 level ranked in the upper 50% of all participants) after treatment, while *Veillonella parvula*, and *Prevotella copri* were associated with a low response in GLP-1 (i.e., the percentage change in GLP-1 level ranked in the lower 50% of all participants) after treatment; however, all associations were not statistically significant (*p* > 0.05).

##### SGLT2 Inhibitor and Sulfonylureas

One study [26], with the RCTs of two drugs, SGLT2 inhibitors and sulfonylureas, reported that baseline microbiome composition did not predict treatment-induced metabolic changes (i.e., changes in glycaemic control, fasting insulin, body mass index, fat mass percentage, and waist circumference) after a 12-week treatment either in dapagliflozin (SGLT2 inhibitor) (*n* = 24) or in gliclazide (sulfonylureas) (*n* = 20) (*p* > 0.05). This study also showed no significant associations between the microbiome profile and clinical parameters, including glycaemic control, fasting insulin, etc., suggesting that the observed metabolic changes are unlikely to be mediated by the gut microbiome.

## 4. Discussion

We performed a systematic review weighing the evidence from observational and RCTs on the (bidirectional) associations of glucose-lowering medication and gut microbiome profiles in T2D patients. This is the first systematic review summarising existing evidence on the effects of glucose-lowering drugs on the gut microbiome and the gut microbiome’s influence on glycemic control in T2D patients.

### 4.1. Effects of Glucose-Lowering Drugs on Gut Microbiome Composition

Overall, the findings from different studies showed variations concerning the effects of glucose-lowering medications on the gut microbiome. Variation across different studies might be explained by differences in sample collection and storage methods, bioinformatic pre-processing pipeline and used genome reference for taxa mapping, thresholds for low-quality reads filtering, normalisation methods, differential abundance taxa detection methods, and adjusted covariates [41]. For all RCTs, a causal relationship can be derived: gut microbiome changes are the causal effect of glucose-lowering medication use. For observational studies, as there is a confounding risk that might explain significant relationships between variables unrelated by cause and effect [40], it can only be established that significant associations exist between glucose-lowering medication use and gut microbiome changes.

#### 4.1.1. Metformin

As a first-line glucose-lowering drug, metformin is widely used in T2D patients. Overall, observational studies [15,16,17,18,19,20,21] reported 13 increased taxa and 23 decreased taxa associated with metformin treatment. Whereas, RCTs [22,23] reported 12 increased taxa and six decreased taxa, among which the negative association of *Intestinibacter* and *Oscillibacter* with metformin was also supported by one of the selected observational studies separately [16,20]. It was reported that a higher abundance of *Intestinibacter* is associated with an increased risk of Crohn’s disease [42] and sleep problems [43]. This suggested that metformin medication could potentially benefit many relevant disorders. *Oscillibacter* was reported to be enriched in T2D [44] and also negatively associated with intestinal barrier function [45]. In addition, one experimental study showed that *Oscillibacter* was a potentially important gut microbe that mediated a high-fat diet-induced gut dysfunction in mice [46]. This indicated that one of the functional mechanisms of metformin might be maintaining the gut barrier integrity. In summary, studies evaluating metformin reported a broad spectrum of changed taxa after metformin treatment, among which different studies highlighted the role of *Intestinibacter* and *Oscillibacter.*

#### 4.1.2. Other Drugs

For α-GI, four studies [15,24,29,30] reported 16 higher taxonomic taxa and 30 species that increased after α-GI use, and 23 higher taxonomic taxa, as well as 46 species, that decreased after α-GI use. Among the enriched taxa list, positive associations of *Bifidobacterium* and *Lactobacillus* were supported by all four studies; and *Megasphaera* was supported by two [29,30] of the four studies. Notably, *Bifidobacterium* and *Megasphaera* were also reported to be positively associated with the metformin treatment [17,20,22], highlighting the shared role of these taxa in the therapeutic effects of these drugs. *Bifidobacterium* is known for its beneficial health effects, including regulation of intestinal microbial homeostasis, modulation of local and systemic immune responses, inhibition of pathogens and harmful bacteria [47], improvement of the gut mucosal barrier, and lowering lipopolysaccharide levels in the intestine [48]. In addition, *Bifidobacterium* species can improve glucose homeostasis by enhancing expressions of insulin signalling proteins and improving the adipokine profile in diabetic mice [49]. *Megasphaera* is a gut microbe that can produce SCFAs, such as acetate, propionate, butyrate, and valerate [50]. *Lactobacillus* is a tryptophan metabolite-producing bacteria [51] and can restore insulin sensitivity and improve glucose metabolism [52]. A recent study [53] showed that some strains of *Lactobacillus* can significantly alleviate blood glucose in T2D mice. Previous evidence [1] has hypothesised that the effects of *Lactobacillus* on glucose homeostasis and insulin sensitivity come from the bacteria-mediated butyrate production that acts as a metabolic modulator. Moreover, *Bifidobacterium* and *Lactobacillus* can promote the production of GLP-1 [54]. Among the taxa that decreased after α-GI treatment, negative associations of *Bacteroides* and *Clostridium* were supported by three [24,29,30] of the four studies; and *Alistipes* was supported by two [24,30]. Based on a previous study, *Bacteroides* plays divergent roles in either protecting against or promoting infection [55] and was reported to contain important pathogens [56]. In addition, *Alistipes* and *Clostridium* were also reported as negatively associated with metformin treatment [20], suggesting that α-GI and metformin might have common effects on the gut microbiome. *Alistipes* were reported to be associated with inflammation, cancer, and mental health; whether their role is positive or negative remained conflicting in different studies [57]. These findings indicate that, like metformin, α-GI treatment could improve the gut microbiome ecosystem by increasing the proportion of healthy gut microbes.

Regarding the GLP-1RA, only one observational study [25] investigated its relationship with gut microbiome profiles, finding 15 taxa with increased abundance and 20 taxa with decreased abundance after treatment. Among the reported decreased taxa, *Oscillospira* and *Alistipes* were negatively associated with the metformin treatment [17,22]. *Alistipes* have been discussed above and *Oscillospira* is postulated as a candidate for next-generation probiotics because of its potential beneficial effects on specific metabolic conditions such as obesity [58]. Regarding the DPP-4 inhibitor, the changed gut microbiome profiles were proven to be highly overlapped with other drugs. These included increased *Bifidobacterium* and decreased *Fusobacterium*, *Parabacteroides*, and *Bacteroides.*

Evidence is uncertain regarding the influence of glucose-lowering medication on microbiome α diversity, as findings from the included studies in this systematic review were inconsistent. Overall, there is strong evidence around the contention that treatment with glucose-lowering compounds can increase in T2D patients the abundance of beneficial bacteria (such as SCFA producers) and/or decrease the abundance of harmful bacteria (taxa that accumulated in pathological conditions or established as pathogens).

### 4.2. Effects of the Gut Microbiome on Glycemic Control

The mechanisms of the gut microbiome influencing glycaemic control are complex. Potential mechanisms whereby gut dysbiosis contributes to metabolic dysfunction include microbiota-driven increases in systemic lipopolysaccharide concentrations, changes in bile acid metabolism, alterations in short-chain fatty acid production, alterations in gut hormone secretion, and changes in circulating branched-chain amino acids, as described previously [59]. In addition, the interaction between the gut microbiome and intestinal immunity, the microbiota-gut-brain axis, and the microbiota-gut-liver axis are also involved in the regulating process of glucose [60].

In our review, two studies (one observational study and one RCT) [20,22] found that the effects of the gut microbiome on glycemic control were associated with metformin treatment. The observational study [20] highlighted that the baseline abundance of *Prevotella copri* was positively associated with ineffective treatment response and the abundance of *E. faecium*, *Lactococcus lactis*, *Odoribacter*, and *Dialister* were all positively associated with an optimal glycemic response to the metformin treatment. The RCT [22] revealed abundance of five taxa, including *Bacteroides*, *Parabacteroides*, *Alistipes*, *Oscillibacter*, and *Ruminococcaceae* that decreased after metformin treatment were associated with poor glycemic control, and the abundance of 12 taxa (see Table 3) that increased after metformin treatment were significantly correlated with adequate glycemic control. Among effective response-related taxa, *E. faecium* was reported to be related to body weight loss, reduction of serum lipid levels and blood glucose levels, and improved insulin resistance in experimental rats fed with a high-fat diet [61]. *Lactococcus lactis* was reported to positively influence hyperglycaemia reduction, glucose tolerance improvement, and insulin secretion enhancement [62,63]. *Odoribacter* was negatively correlated with insulin resistance based on a study in non-diabetic Japanese men [64]. *Dialister* is a taxon that possibly mediates the beneficial effects of whole-grain diets on improved metabolic health based on clinical trials [65]. *Blautia* is a SCFAs producer associated with improved glucose and lipid homeostasis [66]. In addition, one study showed that the oral administration of *B. wexlerae* (a *Blautia* species) in mice could induce metabolic changes and anti-inflammatory effects by producing metabolites, such as S-adenosylmethionine, acetylcholine, L-ornithine succinate, lactate, and acetate [67]. *Anaerostipes* are butyrate producers [68] associated with reduced T2D risk [69]. *Among the taxa* that were negatively correlated with glycemic control, *Prevotella copri* was reported to be associated with an increased risk of rheumatoid arthritis [70]. The other five taxa, including *Bacteroides*, *Parabacteroides*, *Alistipes*, *Oscillibacter*, and *Ruminococcaceae* were reported as pathogenic bacteria [12].

Overall, the findings of these two studies suggest that specific taxa are associated with the glycemic response to metformin treatment and might mediate the therapeutic effects of metformin. Despite this, the cause-and-effect relationship between glycemic response and specific taxa abundance changes cannot be established based on studies included in this review. This is because in the previous RCT [22] both glycemic response and microbiome composition changes were after treatment, and it was unclear about the time sequence of their occurrence. Based on the literature, supplementation of *Lactobacillus* species [71] or other probiotics [72] could improve glycemic parameters in T2D patients, which helps establish the evidence of a cause-and-effect relationship between gut microbiome changes and glycemic control.

### 4.3. Summary and Conclusions

To our knowledge, this is the first systematic review that evaluated evidence from clinical studies and trials that investigated the bi-directional relationship between glucose-lowering medications and the gut microbiome in T2D patients. Our review presented comprehensive evidence on the interaction between the gut microbiome and glucose-lowering drugs in T2D patients receiving medication treatment. We also evaluated the degree of uncertainty on every point of evidence presented in this review from an epidemiology perspective. We highlighted the cause-and-effect relationship between some glucose-lowering medications and specific gut microbiome taxa changes based on evidence from all RCTs. We pinpointed the level of uncertainty and caution needed when interpreting results between glucose-lowering medications, microbiome diversity, and glycemic control.

Although glucose-lowering drugs might influence the gut microbiome through different mechanisms, a common influence on gut microbiome shared by these drugs seems to be promoting SCFAs-producing bacteria that can induce insulin sensitivity enhancement, improve energy metabolism, attenuate systemic inflammation, and inhibit harmful bacteria, such as pathogens. In line with this, the past literature reported that increased post-intervention SCFA is associated with lower fasting insulin, which benefits insulin sensitivity based on findings from a systematic review and meta-analysis of clinical studies and trials [73]. This added extended evidence on the role of SCFA-producing bacteria in diabetes.

In summary, this systematic review highlighted that these medications could, on the one hand, increase taxa that can enhance the host immune system or restore insulin sensitivity, such as *Bifidobacterium* spp., *Lactobacillus*, *Megasphaera*, *Blautia*, and *Anaerostipes*; and on the other hand, decrease harmful taxa, such as *Bacteroides*, *Intestinibacter*, *Oscillibacter*, *Alistipes*, and *Clostridium*. Our results also showed that T2D patients with poor compared to optimal glycemic response to medication treatment presented a significantly different abundance of gut microbiome taxa, suggesting that specific bacterial taxa might mediate metformin’s therapeutic effects.

As evidence regarding the influence of glucose-lowering medication on microbiome α diversity and α diversity on glycemic control was uncertain, further high-quality research is required to determine the bi-directional influence between glucose-lowering medication and microbiome diversity.

## 5. Limitations, Clinical Implications, and Future Perspectives

Our work is not free of limitations. First, we only included published studies written in English; therefore, our work could have omitted high-quality studies on that subject written in other languages. Second, given the included studies’ different methodologies and analytical pipelines, a meta-analysis could not be performed. Third, due to the heterogeneity of these studies involved in this review, we were unable to analyse the effects of drug treatment duration on GM composition. As longer follow-ups may affect the gut microbiome structure differently compared to shorter follow-ups, this could potentially bias the conclusion. Even though, for the first time, our systematic review presented comprehensive evidence on the interaction between the gut microbiome and glucose-lowering drugs on T2D patients receiving medication treatment. Moreover, the degree of uncertainty on every point of evidence presented in this review was also evaluated from an epidemiology perspective. In addition, as shown in the literature search protocol of our review, the number of RCTs presented is limited compared to observational studies. Therefore, we propose more RCTs should be designed and performed to study the interactions between gut microbiome and glucose-lowering drugs in the future.

As the gut microbiome is easily influenced by outside environmental factors [8], a more comprehensive approach, including simultaneously the contribution of gut microbes and other host-related (e.g., genetics and metabolomics) and environmental/lifestyle factors, is still needed in future studies. This integration step is critical to understanding the effect of the gut microbiome in the individualised therapy of T2D. Considering the vast inter-individual variability of gut microbiome profile, using standardised methods, and applying novel machine learning techniques to cluster gut microbiome before and after interventions is necessary to help translate research outputs into optimised precision therapy in real-world clinical settings.

## Figures and Tables

**Figure 1 genes-14-01572-f001:**
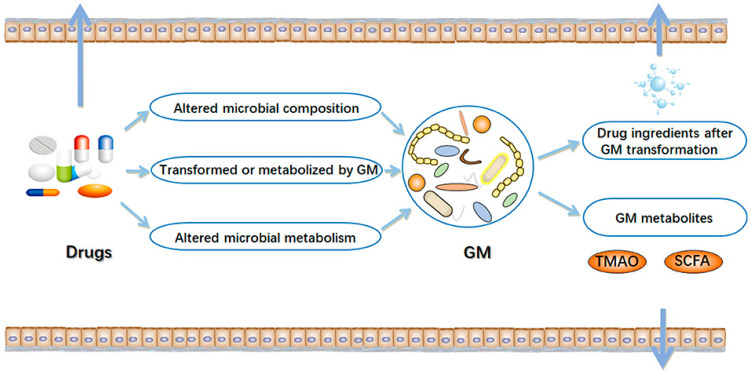
Interactions between drugs and gut microbiome (GM) [13]. In the intestinal tract, there are complex interactions between drugs and microorganisms. On the one hand, drugs can result in alterations in the composition and function of the gut microbiome. On the other hand, the gut microbiome may alter the chemical structure of drugs and directly or indirectly affect drug efficacy. TMAO, trimethylamine N-oxide; SCFA, short-chain fatty acids. The arrow mark indicates drugs, GM metabolites and drug ingredients after GM transformation are transferred outside the gut.

**Figure 2 genes-14-01572-f002:**
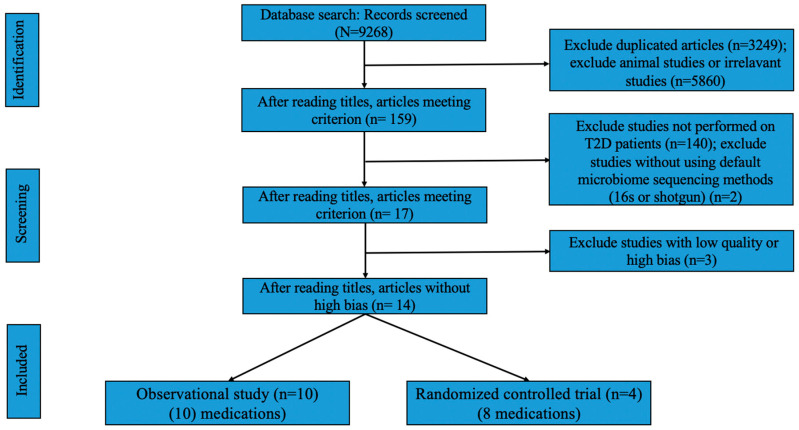
PRISMA flow diagram detailing the inclusion process.

**Table 1 genes-14-01572-t001:** General view of included studies and risk-of-bias evaluation.

Source [Reference]	Study Type	Intervention Group (Sample Size)	Control Group (Sample Size)	Follow-Up	R	D	Mi	Me	S	Co	O
Zhang et al. [17]	Observational study	T2D patients with metformin treatment (n = 51)	T2D patients without metformin treatment (n = 26)	>3 months	-	-					
Forslund et al. [18]	Observational study	T2D patients with metformin treatment (n = 93)	T2D patients without metformin treatment (n = 106)	NA	-	-					
Cuesta-Zuluaga et al. [19]	Observational study	T2D patients with metformin treatment (n = 14)	T2D patients without metformin treatment (n = 14)	NA	-	-					
Sun et al. [20]	Observational study	T2D patients with metformin treatment (n = 30)	The same patients before treatment	3 days	-	-					
Napolitano et al. [21]	Observational study	T2D patients with metformin treatment (n = 14)	The same patients before treatment	3 months	-	-					
Elbere et al. [14]	Observational study	T2D patients with metformin treatment (n = 50)	The same patients before treatment	7 days	-	-					
Nakajima et al. [22]	Observational study	T2D patients with metformin treatment (n = 21)	The same patients before treatment	4 weeks	-	-					
Zhang et al. [17]	Observational study	T2D patients with α-GI treatment (n = 17)	T2D patients without α-GI treatment (n = 26)	>3 months	-	-					
Shang et al. [25]	Observational study	T2D patients with GLP-1RA (Liraglutide) treatment (n = 40)	The same patients before treatment	4 months	-	-					
Takewaki et al. [29]	Observational study	T2D patients with α-GI treatment (n = 18)	The same patients before treatment	4 weeks	-	-					
Tong et al. [15]	RCT	T2D patients with metformin treatment(n = 100)	The same patients before treatment	12 weeks							
Wu et al. [23]	RCT	T2D patients with metformin treatment (n = 22)	T2D patients with placebo treatment (n = 18)	4 months							
Gu et al. [24]	RCT	T2D patients with α-GI treatment (n = 51)	The same patients before treatment	3 months							
Gu et al. [24]	RCT	T2D patients with Sulfonylureas treatment (n = 43)	The same patients before treatment	3 months							
Bommel EJM et al. [26]	RCT	T2D patients with Sulfonylureas treatment (n = 20)	The same patients before treatment	12 weeks							
Bommel EJM et al. [26]	RCT	T2D patients with SGLT2 Inhibitor treatment (n = 24)	The same patients before treatment	12 weeks							
Zhang et al. [30]	RCT	T2D patients with α-GI treatment (n = 44)	The same patients before treatment	6 months							
Zhang et al. [30]	RCT	T2D patients with DPP-4 inhibitor treatment (n = 48)	The same patients before treatment	6 months							

R—bias arising from the randomisation process; D—bias due to deviations from intended interventions; Mi—bias due to missing outcome data; Me—bias in the measurement of the outcome; S—bias in the selection of reported results; Co—bias due to potential confounding factors; O—overall bias. Low bias—green; some concerns—yellow; high risk—red.

**Table 2 genes-14-01572-t002:** Altered gut microbiome taxa associated with the treatment of glucose-lowering drugs in T2D patients.

Glucose Lowering Drug	Increased Taxa	Decreased Taxa	α Diversity	Study Design [Reference]
Metformin	Spirochaete and *Turicibacter*	Erysipelotrichi and *Ruminococcus*	No effect	Observational study [17]
	*Escherichia* spp.	*Intestinibacter* spp.	NA	Observational study [18]
	*Prevotella*, *Megasphaera*	Oscillospira, Barnesiellaceae, *Clostridiaceae 02d06*	NA	Observational study [19]
	NA	*Bacteroides fragilis (B. fragilis)*	↓	Observational study [20]
	o__Bifidobacteriales, f__Bifidobacteriaceae, *g__Bifidobacterium*, *s__Bifidobacterium_adolescentis*, *t__Bifidobacterium_adolescentis_unclassified*, *g__Barnesiella*, *s__Barnesiella_intestinihominis*, *s__Clostridium_bartlettii*	f__Clostridiaceae, *g__Lactococcus*, *g__Clostridium*, *s__Parabacteroides_distasonis*, *t__Parabacteroides_distasonis_unclassified*, *s__Lactococcus_lactis*, *t__Lactococcus_lactis_unclassified*, *f__Oscillospiraceae*, *g__Oscillibacter*, *s__Oscillibacter_unclassified*, *s__Enterococcus_faecium*, *t__Enterococcus_faecium_unclassified*, *s__Bacteroides_vulgatus*, *t__Bacteroides_vulgatus_unclassified*, *f__Enterococcaceae*, and *g__Enterococcusbacteriales*	No effect	Observational study [14]
	NA	Firmicutes, Firmicutes/Bacteroidetes ratio	No effect	Observational study [22]
	*Clostridium XIVa*, *Erysipelotrichaceae incertae sedis*, *Escherichia-Shigella*, *Fusobacterium*, *Flavonifractor*, *Clostridium XVIII* and *IV*, *Blautia* spp. and *Anaerostipes*	*Bacteroides*, *Parabacteroides*, *Alistipes*, *Oscillibacter*, and *un-Ruminococcaceae*	↑	RCT [15]
	γ-Proteobacteria, *Escherichia coli*, and Firmicutes	*Intestinibacter*	NA	RCT [23]
α-glucosidase inhibitor	*Bifidobacterium* and *Lactobacillus*	NA	No effect	Observational study [15]
	Actinobacteria, *Bifidobacterium*, *Eubacterium*, *Megasphaera*, and *Lactobacillus*	*Bacteroidetes*, *Bacteroides*, *Blautia*, *Prevotella*, *Clostridium*, *Phascolarctobacterium*, and *Lachnoclostridium*	No effect	Observational study [29]
	*Lactobacillus* and *Bifidobacterium*	*Bacteroides*, *Alistipes* and *Clostridium*	↓	RCT [24]
	*Bifidobacterium*, *Lactobacillus*, *Solobacterium*, *Streptococcus*, *Actinomyces*, *Acidaminococcus*, *Megasphaera*, *Veillonella*, *Haemophilus*, *Granulicatella*, *Collinsella*, *Gemella*, *Anaerostipes*, *Rothia*, *Enterococcus*, and 30 species	*Bacteroides*, *Roseburia*, *Alistipes*, *Bilophila*, *Oscillibacter*, *Parabacteroides*, *Clostridium*, *Odoribacter*, *Holdemania*, *Adlercreutzia*, *Barnesiella*, *Flavonifractor*, *Subdoligranulum*, *Ruminococcus*, *Oxalobacter*, *Parasutterella*, *Anaerotruncus*, *Akkermansia*, and 46 species	↓	RCT [30]
Liraglutide/ GLP-1RA	*Streptococcaceae*, *Bacilli*, *Verrucomicrobia*, *Coriobacteriia*, *Coriobacteriaceae*, *Collinsella*, *Akkermansia*, *Verrucomicrobiaceae*, *Coriobacteriales*, *Lactobacillales*, *Verrucomicrobiae*, *Clostridium*, *Clostridiaceae*, *Verrucomicrobiales*, *Actinobacteria*	*Acinetobacter*, *Oscillospira*, *Desulfovibrionales*, *S24_7*, *Fusobacteriaceae*, *Rikenellaceae*, *Pseudomonadaceae*, *Pseudomonadales*, *Desulfovibrionaceae*, *Acidaminococcus*, *Fusobacteriales*, *Succinatimonas*, *Deltaproteobacteria*, *Fusobacteriia*, *Moraxellaceae*, *Megamonas*, *Alistipes*, *Fusobacteria*, *Fusobacterium*, *Megasphaera*	↓	Observational study [25]
Sulfonylu-reas	No significant changes		No effect	RCT [24,26]
SGLT2 Inhibitor	No significant changes		No effect	RCT [26]
Dipeptidyl peptidase 4 inhibitor	*Bifidobacterium*, and 2 species (*Clostridium bartlettii*, and *Bifidobacterium adolescentis*)	*Paraprevotella*, *Fusobacterium*, *Parabacteroides*, *Bacteroides*, and 8 species	No effect	RCT [30]

Taxa names in italics denote taxa are at genus level or lower. “↓”: decreased; “↑”: increased. “NA”: no data available.

**Table 3 genes-14-01572-t003:** Effects of the gut microbiome on glycemic control of glucose-lowering drugs.

Glucose Lowering Drug	Taxa Positively Correlate with Glycemic Control	Taxa Negatively Correlate with Glycemic Control	If α Diversity Influences Glycemic Control	Study Design [Reference]
Metformin	*E. faecium*, *Lactococcus lactis*, *Odoribacter*, *and Dialister*	*Prevotella copri*	No	Observational study [14]
	*Blautia*, *Anaerostipes*, *Clostridium XIVa*, *Erysipelotrichaceae incertae sedis*, *Escherichia-Shigella*, *Fusobacterium*, *Flavonifractor*, *Lachnospiraceae*, *Lachnospiracea incertae sedis*, *Clostridium XVIII and IV*	*Bacteroides*, *Parabacteroides*, *Alistipes*, *Oscillibacter*, *and un-Ruminococcaceae*	NA	RCT [15]
SGLT2 Inhibitor	Not associated	Not associated	NA	RCT [26]
Sulfonylureas	Not associated	Not associated	NA	RCT [26]
α-glucosidase inhibitor	Not significant	Not significant	NA	RCT [30]
Dipeptidyl peptidase 4 inhibitor	Not significant	Not significant	NA	RCT [30]

Taxa names in italics denote taxa are at genus level or lower. “NA”: no data available.

## Data Availability

Data sharing is not applicable. No new data were created or analysed in this study. Data sharing does not apply to this article.

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
