# Peer review of "Bi-Directional Interactions between Glucose-Lowering Medications and Gut Microbiome in Patients with Type 2 Diabetes Mellitus: A Systematic Review"

_genes, 2023, doi:10.3390/genes14081572_

Round 1
Reviewer 1 Report
‘’Bi-directional interactions between glucose-lowering medications and gut microbiome in patients with type 2 diabetes mellitus: A systematic review’’
The manuscript presented for review consists of 16 pages with 70 references.
-References: Most of them are recent. They relate to the whole context of the study. I would suggest to add some studies from 2023 to complete the review.
3 tables and 1 figure are included.
English language is understandable. The style is proper.
The work fits the journal scope - Special Issue ‘’Omics Studies of Type 2 Diabetes and Diabetes-Related Complications’’.
The manuscript is divided into 5 sections (Introductio; Methods; Results; Discussion; Limitations clinical implications and future perspectives). The manuscript is well-structured.
- I would recommend to create a figure which shows potential mechanisms and interactions between glucose-lowering medications and gut microbiome to better visualize the issue.
-By what mechanisms/signaling pathways do gut microbiome affect the regulation of the glucose?
The topic is interesting and valuable for clinical practice.
Author Response
Point 1: Most of them are recent. They relate to the whole context of the study. I would suggest to add some studies from 2023 to complete the review.
Response 1: We thank the reviewer for complementing our literature search. We have systematically updated the reference used up to July 2023 following our protocol and PRISMA workflow. In the revised manuscript, the sections on methods, results and discussion were all updated after incorporating new studies.
Point 2: I would recommend to create a figure which shows potential mechanisms and interactions between glucose-lowering medications and gut microbiome to better visualize the issue.
Response 2: In the revised manuscript, we have added the figure (Figure 1) in the introduction section. Also, in the third paragraph of the introduction section, we have added these sentences to elucidate this point.
“On the one hand, enzyme activity in gut microbial communities can alter the structure of drugs, thereby influencing their bioavailability. Particularly, variability in the component and metabolic capacity of the gut microbiome can significantly influence the toxicity and clinical efficacy of drugs. On the other hand, drugs can also alter the composition and function of the gut microbial community, thereby changing the intestinal microenvironment and affecting microbial metabolism. The interactions between gut microbiome and drugs are summarized in Figure 1, as published previously [13]. ”
Point 3: By what mechanisms/signaling pathways do gut microbiome affect the regulation of the glucose?
Response 3: In the revised manuscript, we have added one paragraph in the discussion section 4.2 to explain this.
“The mechanisms of gut microbiome influencing glycaemic control are complex. Potential mechanisms whereby gut dysbiosis contributes to metabolic dysfunction include microbiota-driven increases in systemic lipopolysaccharide concentrations, changes in bile acid metabolism, alterations in short-chain fatty acid production, alterations in gut hormone secretion, and changes in circulating branched-chain amino acids, as described previously [59]. Besides, the interaction between the gut microbiome and intestinal immunity, the microbiota-gut-brain axis and the microbiota-gut-liver axis are also involved in the regulating process of the glucose [60].”
Reviewer 2 Report
The authors reviewed published observational studies and randomized controlled trials (RCTs) regarding the interactions between glucose-lowering drugs and gut microbiome. The manuscript is well-written, and the topic is interesting in the field.
Below are some concerns I have.
1. The follow-up durations in the observational studies and RCTs included in this review are quite different, how would this variation affect the conclusion?
2. The number of RCTs included in this review seems quite limited.
Quality of English Language is good.
Author Response
Point 1: The follow-up durations in the observational studies and RCTs included in this review are quite different, how would this variation affect the conclusion?
Response 1: This is a good point, and we thank the reviewer for pointing this out. The follow-up durations in these studies varied considerably. Due to the heterogeneity of these studies, we were unable to analyze the effect of drug treatment duration.
We have addressed this limitation in the revised manuscript, section 5, by adding the following sentence “Third, due to the heterogeneity of these studies involved in this review, we were unable to analyse the effect of drug treatment duration on GM composition. As longer follow-up may affect the gut microbiome structure differently compared to shorter follow-up, this could potentially bias the conclusion.”
Point 2: The number of RCTs included in this review seems quite limited.
Response 2: Since this is a systematic review, all included studies were screened and determined following a systematic protocol. In this sense, we have added all RCTs that were eligible for our review. Besides, in section 5, we added the following sentences: “Besides, as shown in the literature search protocol of our review, the number of RCTs presented is a bit limited compared to observational studies. Therefore, we propose more RCTs should be designed and performed to study the interactions between gut microbiome and glucose-lowering drugs in future.”